# Fault Diagnosis of Rolling Bearings Based on WPE by Wavelet Decomposition and ELM

**DOI:** 10.3390/e24101423

**Published:** 2022-10-06

**Authors:** Caiping Xi, Zhibo Gao

**Affiliations:** 1College of Automation, Jiangsu University of Science and Technology, Zhenjiang 212100, China; 2Ocean College, Jiangsu University of Science and Technology, Zhenjiang 212100, China

**Keywords:** wavelet decomposition, weighted permutation entropy, extreme learning machine, rolling bearing vibration signals

## Abstract

The fault diagnosis classification method based on wavelet decomposition and weighted permutation entropy (WPE) by the extreme learning machine (ELM) is proposed to address the complexity and non-smoothness of rolling bearing vibration signals. The wavelet decomposition based on ‘db3’ is used to decompose the signal into four layers and extract the approximate and detailed components, respectively. Then, the WPE values of the approximate (CA) and detailed (CD) components of each layer are calculated and composed to be the feature vectors, which are finally fed into the extreme learning machine with optimal parameters for classification. The comparative study of the simulations based on WPE and permutation entropy (PE) shows that the classification method of seven kinds of signals of normal bearing signals and six types of fault states (7 mils and 14 mils) based on WPE (CA, CD) with the number of nodes in the hidden layers of ELM determined by the five-fold cross-validation has the best performances, the training accuracy can reach 100%, and the testing accuracy can reach 98.57% with 37 nodes of the hidden layer by ELM. The proposed method using WPE (CA, CD) by ELM provides guidance for the multi-classification of normal bearing signals.

## 1. Introduction

As an important part of rotating machinery and equipment, rolling bearings directly affect the running state and service life of machinery and equipment. Effectively extracting features from these signals and making a diagnosis is the key to early fault warning.

Bearing vibration signals contain much information about the motion state of the bearing, so the study of bearing signals for fault diagnosis by the analysis of vibration signals remains the most effective and commonly used method. The faulty rolling bearing vibration signals are time-varying and non-stationary. Weak fault characteristic information is easily submerged by noise, making fault diagnosis difficult. Traditional analysis methods are based on smooth and linear signals. However, in practical applications, vibration signals often behave non-smoothly and non-linearly, so further research on non-smooth signal analysis and extraction methods of non-linear features is needed. Permutation entropy (PE), as an index of sequence complexity, has been widely used in time series complexity and dynamics analysis. Bruzzo A. [1] used the permutation entropy algorithm to detect vigilance changes and normal states in the electrocardiogram (ECG) of epileptic patients with good results. For the fault signals of rolling bearings, the intrinsic dynamics of different fault types are different, which results in different signal complexity. Therefore, the PE algorithm can be used for feature extraction of fault signals. Yan R. [2] demonstrated that the PE algorithm can well discriminate the classes of bearing signals in different states. Permutation entropy does not represent the characteristics of the signal well in some abnormal mutations of the signal, so some improvements are needed to fully reflect the detailed characteristics of the time series.

Common analysis methods include empirical mode decomposition [3] and wavelet decomposition. Polygiannakis J. used continuous wavelet transform to decompose the unique time series and apply it to the sunspot index [4]. Griffiths K.R. used wavelet analysis to decompose the vibration simulation signal caused by roads in order to improve the pre-distribution mode of packaging [5]. According to the actual non-stationary vibration signals of large equipment in a large iron and steel company, Fan X.B. et al. conducted a time-frequency analysis using wavelet packet decomposition and reconstruction methods. They analyzed the influence of wavelet decomposition on signal denoising and the influence of the selection of high-frequency weight coefficients at each layer on signal denoising [6]. The empirical mode decomposition method can be used to analyze complex signals adaptively without relying on pre-selected basis functions and has a large number of applications in the analysis of non-linear and non-stationary signals. Wavelet decomposition inherits and develops the idea of localization of the short-time Fourier transform and also overcomes the disadvantage that the window size cannot vary with frequency. It provides a ‘time-frequency’ window that varies with frequency. In the area of pattern recognition, the main methods include back propagation (BP) neural networks [7], support vector machines [8], and deep learning [9]. BP neural network algorithms are slow to converge, whereas deep learning requires a large number of samples and takes too long to train. ELM is a kind of feed-forward neural network with a single hidden layer, which has the characteristics of fast operation speed and good generalization performance.

In summary, the selection of suitable feature extraction methods and diagnostic models is important for the fault diagnosis of rolling bearing signals. In view of the complexity and non-smooth characteristics of the bearing signals, this paper proposes a fault diagnosis method based on wavelet decomposition and weighted permutation entropy (WPE) [10,11,12] by ELM. The wavelet decomposition of four layers based on ‘db3’ is used to decompose the bearing signals, and the WPE value is calculated for each layer of the decomposed signals, and the results are input into the ELM model as a feature matrix for diagnosis and classification. The experimental results show that the ELM has high accuracy on the entropy features extracted from the approximate component and the detail component after wavelet decomposition.

This paper is organized as follows. Section 2 describes the PE algorithm, WPE algorithm [13], and wavelet decomposition method. Section 3 introduces the specific steps of the proposed classification algorithm based on PE and WPE by ELM [14]. Section 4 describes the data used in this paper, the selection of the parameters of entropy algorithms and ELM, presents the results of the numerical experiments based on the proposed methods of WPE and PE, and compares the results of the two methods. Section 5 summaries the main findings and conclusions of the article.

## 2. Methodology

### 2.1. Permutation Entropy Algorithm

Let the time series X={x(n),n=1,2,…,N} of length N be reconstructed in phase space for the time series. Obtain the matrix $XX_{K}$ of N−(m−1)τ rows and m columns:(1)$$\left[\begin{array}{cccc} x(1) & x(1+\tau ) & \cdots & x(1+(m-1)\tau )\\ \vdots & \vdots & & \vdots \\ x(i) & x(i+\tau ) & \cdots & x(i+(m-1)\tau )\\ \vdots & \vdots & & \vdots \\ x(K) & x(K+\tau ) & \cdots & x(K+(m-1)\tau ) \end{array}\right],i=1,2,\cdots K$$
where K=N−(m−1)τ, m is the embedding dimension and τ is the time delay. Each row in the matrix $XX_{K}$. is a reconstructed vector and there are K vectors. Rearranging the i-th reconstructed vector $X_{i}=[x(i),x(i+\tau ),\cdots ,x(i+(m-1)\tau )]$ in ascending order gives:(2)$$x(i+(j_{1}(i)-1)\tau )\leq x(i+(j_{2}(i)-1)\tau )\leq \cdots \leq x(i+(j_{m}(i)-1)\tau )$$
where i=1,2,⋯,K, $j_{1}(i),j_{2}(i),\cdots ,j_{m}(i)$ indicates the index of the reconstructed vector in which the element is located. If the elements are equal, they are arranged according to the size of the element index. Thus, each reconstructed component of the matrix $XX_{K}$ is rearranged in ascending order to give a matrix of symbolic sequences of K rows and m columns. Denote $\pi_{i}=(j_{1}(i),j_{2}(i),\cdots ,j_{m}(i)),i=1,2,\cdots ,K$, $\pi_{i}$ as one of the permutations of (1,2,⋯,m) and the elements of having at most m! different permutations, denoted as ∏.

Use $\pi_{r}$ to denote some permutation in $\pi_{i}$ where 0<r≤m!, count all permutations of $\pi_{i}$ and find the probability of occurrence of each $\pi_{r}$ by the equation:(3)$$p(\pi_{r})=\frac{\sum_{j=1}^{K}l_{A:type(u)=\pi_{r}}(X_{j})}{\sum_{j=1}^{K}l_{A:type(u)\in \prod }(X_{j})}$$
where type(·) denotes the map from pattern space to symbol space, u denotes reconstructed vector $X_{j}$,$l_{A}(u)$ denotes the indicator function of set A (A denotes the set of all the reconstructed vectors belonged to the same type $\pi_{r}$) defined as $l_{A}(u)=\left\{\begin{array}{c} 1,u\in A\\ 0,u\notin A \end{array}\right.,\prod ={\left\{\pi_{r}\right\}}_{r=1}^{m!}$. The final permutation entropy expression is defined as:(4)$$PE(m)=-\sum_{r=1}^{m!}p(\pi_{r})\log_{2}(p(\pi_{r}))$$

When we do the simulation, we obtain the normalization of PE(m) by equation $PE(m)/\log_{2}(m!)$.

### 2.2. Weighted Permutation Entropy Algorithm

The rolling bearing vibration signals may have mutations, the same ordering but with particularly large differences in magnitude and the permutation entropy ignores the effect of such mutations and does not take into account the effect of magnitude on the results. For example, the vectors [3,5,19] and [3,5,100] are both sorted as a permutation of [0,1,2]. However, the difference between 5 and 19 in the vector [3,5,19] is significantly smaller than the that between 5 and 100. Therefore, the effect of signal mutation is added to the calculation of the probability of occurrence for each permutation pattern by adding the appropriate weights to the calculation of the probability of occurrence for each permutation:(5)$$p_{\omega }(\pi_{r})=\frac{\sum_{j=1}^{K}\left[\omega_{r}\cdot l_{u:type(u)=\pi_{r}}(X_{j})\right]}{\sum_{j=1}^{K}\left[\omega_{r}\cdot l_{u:type(u)\in \prod }(X_{j})\right]}$$
where type(·) denotes the map from pattern space to symbol space, u denotes reconstructed vector $X_{j}$, $l_{A}(u)$ denotes the indicator function of set A (A denotes the set of all the reconstructed vectors belonged to the same type $\pi_{r}$) defined as $l_{A}(u)=\left\{\begin{array}{c} 1,u\in A\\ 0,u\notin A \end{array}\right.,\prod ={\left\{\pi_{r}\right\}}_{r=1}^{m!}$. $\omega_{r}$ is the weight of the reconstructed vector $X_{j}$, represented by the variance of $X_{j}$:(6)$$\omega_{r}=\frac{1}{m}\sum_{q=1}^{m}{\left[x(j+(q-1)\tau )-{\bar{X}}_{j}\right]}^{2}$$
${\bar{X}}_{j}$ is the mean value of the reconstructed vector $X_{j}$:(7)$${\bar{X}}_{j}=\frac{1}{m}\sum_{q=1}^{m}x(j+(q-1)\tau )$$

Considering the effect brought about by mutations in the time series, weighted permutation entropy (WPE) is a refinement of permutation entropy. Thus, the WPE expression is defined as:(8)$$WPE(m)=-\sum_{r=1}^{m!}p_{\omega }(\pi_{r})\log_{2}(p_{\omega }(\pi_{r}))$$

Normalization of WPE(m) gives:(9)$$0\leq WPE(m)=WPE(m)/\log_{2}(m!)\leq 1$$

The weighted permutation entropy (WPE) takes full account of the magnitude information of the reconstructed vector. The WPE can fully reflect the detailed characteristics of the time series.

For the time series $X_{1}=[1,2,3,5,100,10,50]$, with the embedding dimension m=3 and time delay τ=1, we can obtain 5 vectors with 3 consecutive neighbor values to construct a matrix $\left[\begin{array}{ccc} 1 & 2 & 3\\ 2 & 3 & 5\\ 3 & 5 & 100\\ 5 & 100 & 10\\ 100 & 10 & 50 \end{array}\right]$. We compare the 3 consecutive values in each row; we find that the 3 vectors [1,2,3], [2,3,5], and [3,5,100] represent the same permutation [0,1,2] since they are in increasing order. The [5,100,10] corresponds to the permutation [0,2,1], for $x_{t}\leq x_{t+2}\leq x_{t+1}$. The [100,10,50] corresponds to the permutation [1,2,0] for $x_{t+1}\leq x_{t+2}\leq x_{t}$. The normalized value of $PE(3)=[-(3/5)\log_{2}(3/5)-(1/5)\log_{2}(1/5)$$-(1/5)\log_{2}(1/5)]/\log_{2}(3!)=0.5304$.

For WPE algorithm 0<r≤m!, we obtain 5 vectors with 3 consecutive values, and for each vector of 3 consecutive values, we can obtain the variance of each vector, then we obtain a weighted vector with 5 values [0.6667, 1.5556, 2048.7, 1905.6, 1355.6]. We then obtain the sum of the variances corresponding to the same permutation, for the permutation [0,1,2] is the sixth permutation vector of m=3 (the first permutation is [2,1,0], the second one is [2,0,1], the third one is [1,2,0], the fourth one is [1,0,2], the fifth one is [0,2,1], the sixth permutation is [0,1,2]), the sum of the variances [0.6667,1.5556,2048.7] is 2050.9 for the sixth permutation. The third permutation [1,2,0] only has one variance 1355.6, the fifth permutation [0,2,1] only has one variance 1905.6, and then we obtain the [1355.6, 1905.6, 2050.9]. We obtain the probability of occurrence for each permutation using [1355.6, 1905.6, 2050.9] divided by 5312.1 (the sum of the three values in the vector [1355.6, 1905.6, 2050.9]), that is [0.2552, 0.3587, 0.3861], for the third permutation [1,2,0], the probability is 0.2552, for the fifth permutation [0,2,1], the probability is 0.3587, for the sixth permutaition [0,1,2], the probability is 0.3861, and then we obtain WPE(3)=0.6048 by $[-(0.2552)\log_{2}(0.2552)-(0.3587)\log_{2}(0.3587)-(0.3861)\log_{2}(0.3861)]/\log_{2}(3!)=0.6048$; the larger the entropy is, the more complex the signal is. Therefore, the WPE(3) value is larger than the PE(3) value, which means the WPE can describe more characteristics of the complexity of mutation in the signal.

For the time series $X_{2}=[1,2,3,5,19,10,50]$, with the embedding dimension m=3 and time delay τ=1, we can obtain 5 vectors with 3 consecutive neighbor values to construct a matrix $\left[\begin{array}{ccc} 1 & 2 & 3\\ 2 & 3 & 5\\ 3 & 5 & 19\\ 5 & 19 & 10\\ 19 & 10 & 50 \end{array}\right]$. Comparing the 3 consecutive values in each row, we find that the 3 vectors [1,2,3], [2,3,5] and [3,5,19] represent the same permutation [0,1,2] since they are in increasing order. The [5,19,10] corresponds to the permutation [0,2,1] for $x_{t}\leq x_{t+2}\leq x_{t+1}$. The [19,10,50] corresponds to the permutation [1,0,2] for $x_{t+1}\leq x_{t}\leq x_{t+2}$. The normalized value of $PE(3)=[-(3/5)\log_{2}(3/5)-(1/5)\log_{2}(1/5)$
$-(1/5)\log_{2}(1/5)]/\log_{2}(3!)=0.5304$. WPE(3)=0.3841. We can see that the PE values are the same for $X_{1}$ and $X_{2}$, but the WPE values are not the same; for $X_{2}=[1,2,3,5,19,10,50]$, the WPE value is apparently smaller than that of $X_{1}=[1,2,3,5,100,10,50]$. It means the time series $X_{2}=[1,2,3,5,19,10,50]$ has relatively slow changes compared with the time series $X_{1}=[1,2,3,5,100,10,50]$.

For the time series $X_{3}=[1,2,3,5,9,10,50]$, with the embedding dimension m=3 and time delay τ=1, PE(3)=0, and WPE(3)=0. We can see that the PE and WPE values are 0 because of the signal with the same increasing order. Figure 1 shows the data waveforms of three signals used for the explanation of PE and WPE.

### 2.3. Wavelet Decomposition Theory

Wavelet decomposition is a commonly used signal processing method. The discrete wavelet transform (DWT) represents the decomposition of the original time domain signal, which gives the approximate (low frequency) component of the original signal and the detailed (high frequency) component. The approximate component reflects more clearly the essential information of the signal and the detailed component reflects the noise. Wavelet decomposition means that the approximate component of the original signal after the DWT transform is then subjected to n−1 DWT transforms, where n denotes the number of decomposition layers.

The scale parameter a and the translation parameter b are discretized. Choosing $a=a_{0}^{m}$, $b=nb_{0}a_{0}^{m}$, and $a_{0}$ to be the scaling steps and $a_{0}\neq 1$, $b_{0}>0$—related to the specific form of the wavelet φ(t), m, n—to be integers, the discrete wavelet is defined as
(10)$$\varphi_{m,n}=\frac{1}{\sqrt{a_{0}^{m}}}\varphi (\frac{t-nb_{0}a_{0}^{m}}{a_{0}^{m}})=a_{0}^{-m/2}\varphi (a_{0}^{-m}t-nb_{0})$$

Then, the discrete wavelet transform can be obtained:(11)$$W_{f}(m,n)=a_{0}^{-m/2}\int_{-\infty }^{+\infty }f(t)\varphi_{m,n}(t)dt=a_{0}^{-m/2}\int_{-\infty }^{+\infty }f(t)\varphi (a_{o}^{-m}t-nb_{0})dt$$

The current common method of discretization in engineering applications is to perform a binary discretization of the scale and offset parameters, i.e., $a_{0}=2$, $b_{0}=1$, and the resulting wavelet is called a binary wavelet:(12)$$\varphi_{m,n}(t)=2^{-m/2}\varphi (2^{-m}t-n)$$

In contrast, the orthogonality of $\varphi_{m,n}(t)$ allows for the elimination of redundant correlations between two points in wavelet space, so that, compared to the continuous wavelet transform, it does not lose essential information and the result of the transform is more reflective of the nature of the signal itself. The wavelet decomposition diagram is schematically shown in Figure 2:

## 3. Fault Diagnosis Method Based on WPE by Wavelet Decomposition and ELM

For the rolling bearing vibration signals with non-linear and non-smooth characteristics, a bearing fault diagnosis and classification method based on wavelet decomposition and weighted permutation entropy as the detection feature by ELM is proposed. The flowchart of the proposed method is shown in Figure 3 To verify the effectiveness of the proposed method, we use the method based on PE by wavelet decomposition and ELM to do the comparative study.

Step 1. Determine the optimal calculation length of the raw data to truncate the data to ensure the fastest calculation accuracy and calculation time.

Step 2. Determine the optimal WPE parameters and select the best values for the WPE parameters by comparing and analyzing the effect of different embedding dimensions *m* and delay times τ on the results.

Step 3. Wavelet decompositions of the used types of signals with different layers based on different wavelet basis functions are performed on the truncated data to calculate the WPE values of the approximate and detailed components of each layer, and the calculated WPE values are used to construct the feature matrix and feed into ELM by the five-fold cross-validation method to determine the decomposition layers and wavelet basis function by the analysis of the best training accuracy (the total number of the correct predictions of all the signals of used types of divided by the number of all the signals of used types).

Step 4. Determine the parameters of the extreme learning machine (ELM) model using the five-fold cross-validation method, i.e., by the range of the number of nodes the hidden layer determined via the best training accuracy of each fold. ELM, which is a new type of single implicit layer feedforward neural network, consisted of the input layer, the hidden layer, and the output layer [15].

Step 5. After selecting the optimal parameters, the fixed training set is used to train the rolling bearing fault diagnosis model, the fixed test set is fed into the trained ELM model for testing, and the test results are then outputted.

## 4. Experimentation and Analysis

### 4.1. Bearing Failure Data

The experimental data were obtained from the Bearing Data Center at Western Reserve University, USA. Bearing damage was treated using single-point damage from electrical discharge machining (EDM). The bearing was a drive end bearing SKF 6205-2RS and an acceleration sensor was placed above the drive end bearing housing to capture the vibration acceleration signal of the failed bearing. The data was collected from multiple sets of data under different conditions. The data was selected from the drive end bearing at a speed of and a sampling frequency of 12 kHz. To verify the effectiveness of this method, four different states of data were selected: normal, inner race, outer race, and rolling element.

Figure 4a shows the experimental platform consisting of a 1.5 kW (2 hp) electric motor (left), a torque transducer/encoder (center), a dynamometer (right), and control electronics (not shown). The rotational speed is 1772 r/min, the sampling frequency is 12 kHz, and the sampling time is 10s. Single point faults were introduced to the test bearings using electro-discharge machining with fault diameters of 7 mils (0.1778 mm), 14 mils (0.3556 mm), 21 mils (0.5334 mm), 28 mils (0.7112 mm), and 40 mils (1.016 mm), 1 mil = 0.001 inches. SKF bearings were used for the 7, 14, and 21 mils diameter faults, and NTN equivalent bearings were used for the 28 mils and 40 mils faults. All experimental data use drive-end acceleration data, which are called vibration acceleration signals. The validity of the proposed method in this paper is verified by selecting the data of the normal state and different fault states. We use the first 2048×50=102,400 data points of the signal of each type. The data is divided into 50 equal segments with 2048 points per segment. In this paper, to compare the effectiveness of the proposed method based on the inner-race fault signals of different levels, we will use the normal signal and signals of inner-race faults of levels I1, I2, and I3. I1 stands for slight inner-race fault diameter of 7 mils, I2 stands for series inner-race fault diameter of 14 mils, and I3 stands for serious inner-race fault diameter of 21 mils.

The waveforms of 0.1s can be seen in Figure 4b. In the figure, ‘Normal’ stands for normal bearing signal, I1 stands for slight inner-race fault diameter of 7 mils, and I2 stands for serious inner-race fault diameter of 14 mils. Similarly, we define that O1 stands for slight outer-race fault diameter of 7 mils, O2 stands for serious outer-race fault diameter of 14 mils, O3 denotes the serious outer-race fault diameter of 21 mils, B1 stands for slight ball fault of 7 mils, B2 stands for slight ball fault diameter of 14 mils, and B3 denotes the serious ball fault diameter of 21 mils. In this paper, we have seven kinds of signals in different states, such as normal bearing signals. The six fault states of data are I1, I2, O1, O2, B1, and B2. The validation acceleration data of each state is divided into 50 sets and we can obtain 350 sets. The time domain waveforms of bearing vibration signals of the seven different states (Normal, I1, I2, O1, O2, B1, B2) are shown in Figure 5. ‘Normal’ stands for normal bearing signal.

### 4.2. Selection of Weighted Permutation Entropy Parameters

In the calculation of the weighted permutation entropy, three parameter values need to be considered, the length of the time series N, the embedding dimension m, and the delay time τ. Different parameter settings will have a certain impact on the calculation results of the entropy value.

Firstly, to verify the effect of the delay time τ on the calculated WPE values, the WPE values are plotted against the number of embedding dimensions at different delay times for a normal bearing vibration signal of length 2048. Figure 6 shows the WPE and PE curves of normal bearing signals with different time delays τ=[1,2,3,4,5,6,7] and different embedding dimensions m=[3,4,5,6,7]. The values are normalized ones. As shown in Figure 6, it can be seen that the WPE value of the bearing signal is minimally affected when the delay time τ is varied at 2~7. Considering the calculation time issue, we use τ=2 for the calculation of WPE in this paper. When τ=1,2, we can see that WPE curves are lower than PE curves, which indicates that the normal signal is more regular and has fewer abrupt changes.

For a *d*-dimensional system, the phase space is guaranteed to accommodate the features of the original state space when m≥2d+1 [16]. If m is too small, the system characteristics are not well represented, whereas when m is too large, the effect of noise is simultaneously amplified as the embedding dimension becomes larger and the computation time of the algorithm is prolonged, which is not beneficial in practical applications. Therefore, the selection of the embedding dimension is particularly important. Cao, Bandt [17,18] suggested that the permutation entropy value can best characterize the dynamic properties of the time series with m=5, m=6 or m=7. Therefore, the embedding dimension of the WPE should also be chosen as m=5, m=6 or m=7.

Figure 7 shows the WPE and PE curves of normal bearing signals with different data lengths and different embedding dimensions (m=[3,4,5,6,7]) with τ=2. The data lengths N are 256, 512, 1024, 2048, and 4096, respectively. As shown in Figure 7, the changes in WPE values with changing lengths are relatively stable when m=3 and m=4. When the embedding dimension m>5, the WPE values of normal bearing signals with several different lengths start to show a significant decrease. Moreover, Figure 8 shows the curves of the sample standard deviations $error=\sqrt{(1/(n-1))\sum_{i=1}^{n}{\left(x_{i}-\frac{1}{n}\sum_{i=1}^{n}x_{i}\right)}^{2}}$ of the WPE and PE values of different lengths with the fixed embedding dimension value m=[3,4,5,6,7]. For the errors of WPE and PE of the normal bearing signals with several different lengths, those with the embedding dimension m=5 are the smallest. The embedding dimension means that a vector of m elements can have at most m! permutation patterns, the more patterns in the rows, the longer the calculation time. Taking into account the computational time and accuracy, the length of the signal sequence is chosen to be 2048 and the embedding dimension m=5 is chosen as the best parameter for the simulations in Section 4.3.

After truncating the data of seven states of the rolling bearing variation signals (Normal, I1, I2, I3, O1, O2, O3, B1, B2, B3) consisted of one kind of normal bearing signal and three damage diameters of three types of faults (inner-race fault, outer-race fault, and rolling ball fault) of the same bearing SKF 6205-2RS, detection parameters need to be obtained to construct the feature matrix. In this paper, wavelet decomposition and weighted permutation entropy are used to extract the features from the sample data.

Table 1 shows the average accuracy of the training set obtained by the ELM with different decomposition layers of different wavelet basis functions for five-fold cross-validation under the PE and WPE algorithms. In Table 1, ‘Basis’ is the wavelet basis function; ‘PE’ and ‘WPE’ denote permutation entropy and weighted permutation entropy algorithms; ‘N’ is the number of wavelet decomposition layers; ‘Positive mean (PM)’ is the average accuracy (the total number of the correct predictions of all the signals of seven types of divided by the number of all the signals of seven types). The average accuracy PM of the training set is obtained by cross-validating a total of 350 groups of normal and six fault signals into five folds, i.e., 50 groups of each fault signal were set into five folds of 10 signals each in turn as the training set for training. For the number of hidden layer nodes K, when K increases from 1 to 100, for each fold we try to find the range of K at the highest training accuracy and finally calculate the union set of the range of K for each fold so as to obtain the range of values of the number of hidden layer nodes K at the highest mean value of detection accuracy. It can be seen that under WPE, the accuracy of the training set is 100%; under PE, the floating range between different decomposition layers is below 1.00% except for N = 8. Considering the scale of wavelet decomposition is too large, it will increase the complexity of data calculation, and if the scale is too small, it will lead to incomplete signal decomposition, so N < 4 is not considered. Therefore, the wavelet of scale $s=2^{4}$ (the number of decomposition layers is four) is selected to decompose the fault signal. The accuracy of the training set under the PE algorithm with different wavelet bases was observed under the premise that the number of decomposition layers is N = 4. The floating range was 0.85%, and the influence of the wavelet basis function on the accuracy is small. The ’db3′ is chosen as the basis function of wavelet decomposition in this paper.

The ‘db3’ wavelet basis function is used for four-layer wavelet decomposition, and then the approximate and detail components decomposed at different scales are extracted, and their weighted permutation entropy can be calculated to construct the feature matrix.

There are 500 samples with a length of 2048 for the 10 signals (Normal, I1, I2, I3, O1, O2, O3, B1, B2, B3) of different damage degrees of the three types of faults (inner-race fault, outer-race fault, and rolling ball fault) of the same bearing SKF 6205-2RS, and the number of samples for each damage level is 50 groups. In order to ensure the accuracy of diagnosis and classification, the data are divided into a training set and test set in proportion; that is, the numbers of samples in the training set and test set for each type of fault signal are 40 and 10, respectively. Finally, the number of samples in the training set is 400, and the number of samples in the test set is 100 for each different fault signal. Different damage degrees of rolling body fault, inner-race fault, and outer-race fault are classified and identified, respectively, and the control signals are normal signals. Then, the different damage degrees of all fault signals are grouped together for classification and recognition.

In this paper, we firstly use 350 samples with a length of 2048 for seven types of signals (Normal, I1, I2, O1, O2, B1, B2) of different damage degrees of inner-race, outer-race, and rolling body faults to determine the number of wavelet decomposition and wavelet basis function. Therefore, we have 280 samples for training and 70 samples for test.

### 4.3. Experimental Results and Analysis

In the process of orthogonal wavelet decomposition, the low-frequency coefficients are generally decomposed into two parts. After decomposition, an approximation coefficient vector (CA) and a detail coefficient vector (CD) are obtained. The information lost in the two consecutive approximation coefficients can be obtained in the detail coefficients. The next step is to further decompose the approximation coefficient vector into two parts, whereas the detail coefficient vector is no longer decomposed. Figure 9 shows the approximation coefficient curves of four-layer wavelet decomposition based on the ‘db3’ of a normal bearing signal. When we use the WPE method to do the calculation by m=5 and τ=2 we can obtain WPE (CA1, CA2, CA3, CA4) = [0.9008, 0.9583,0.8362, 0.8384]. In Figure 9, four pictures from top to bottom correspond to the wavelet decomposition layers 1–4. The amplitude of the waveform of CA1 in the first picture of Figure 9 is relatively small compared with the waveform of CA2 in Figure 9, although there are a lot of fluctuations in the signal, and a lot of changing trends of each embedding vector in the phase space reconstruction matrix with embedding dimension m=5, there are no large abrupt mutations, and the WPE value is 0.9008. It reflects that the time series has a lot of information. Although the two waveforms of CA1 and CA2 are similar to each other at first sight, the fluctuations of the amplitude values become larger in the second picture of CA2 in Figure 9, so the WPE value becomes larger. The waveform of CA3 in Figure 9 has fewer changing trends of data and the WPE is smaller than that of the waveform of CA2 in Figure 9. Compared with the waveforms of CA3 in Figure 9, the waveform of CA4 in Figure 9 misses a lot of small fluctuations, the changing of the magnitudes is not very substantial apparently, so the WPE is similar to that of the waveform of CA3 in Figure 9. The missing information can be found in the waveforms of detail coefficient curves in Figure 10. We can obtain *WPE* (CD1, CD2, CD3, CD4) = [0.7552,0.8009,0.8094, 0.8108] by the same parameters m=5 and τ=2. The magnitudes of the signal of CD1 in Figure 10 are very small, but the maximum magnitude of the signal of CD2 reaches 0.1. Therefore, the WPE value of the CD2 sequence becomes larger than that of CD1 of the normal bearing signal.

At the same time, we can obtain PE (CA1, CA2, CA3, CA4) = [0.9467,0.9776,0.8899,0.8708], PE (CD1, CD2, CD3, CD4) = [0.7715,0.8446, 0.8798,0.8639] by the same parameters m=5 and τ=2. The PE values reflect the information of the permutation types in the signals with no effects on the magnitudes and large mutations in the signals. For simplicity, we use WPE (CA, CD) to denote WPE (CA1, CA2, CA3, CA4, CD1, CD2, CD3, CD4), WPE (CA) to denote WPE (CA1, CA2, CA3, CA4), WPE (CD) to denote WPE (CD1, CD2, CD3, CD4), PE (CA, CD) to denote PE (CA1, CA2, CA3, CA4, CD1, CD2, CD3, CD4), PE (CA) to denote PE (CA1, CA2, CA3, CA4), PE (CD) to denote PE (CD1, CD2, CD3, CD4).

Figure 11 shows the curves of the mean values of WPE and PE of all the CA and CD sequences of four-layer wavelet decomposition of 200 groups of the normal bearing signals and inner-race fault signals of different levels (I1, I2, and I3) by the same parameters m=5 and τ=2. The data set of each type has 50 groups of data with a length of 2048. The horizontal axis labeled as ‘Data index’ denotes the number of the sample data. There are large differences between the curves of the mean values of WPE (WPEmean) of the normal signal and the inner-race fault signal of I3 in Figure 11a. It is difficult to separate the curves of the WPEmean of I2 and the normal signal. In Figure 11b, the curves of the mean values of PE (PEmean) of I1, I2, and the normal bearing signal can be separated from each other, but it is hard to separate the curves of I3 from the normal bearing signal. Therefore, the WPEmean and PEmean parameters will not be used in the simulations of the classification method in this paper.

We focus on the four types of bearing signals; each data set has 50 groups of data with a length of 2048 and we use the first 40 groups of each type to do the training and the last 10 groups to do the testing simulations by ELM. The test data set of each type has 10 groups of data with lengths of 2048. Figure 12 shows the distribution image of the values of WPE and PE of all the CA sequences of four-layer wavelet decomposition of 40 groups of the normal bearing signals and inner-race fault signals of different levels (I1, I2, and I3) by the same parameters m=5 and τ=2 for the test of the fifth fold simulation of five-fold cross-variation. The horizontal axis labeled as ‘Data index’ denotes the number of the sample data. In Figure 12a, we can separate normal bearing signals from I2 and I3, but it is hard to tell normal ones from the I1 signals. In Figure 12b, we can separate normal bearing signals from I3, but it is hard to tell normal ones from the I1 and I2 signals. In Figure 12c, we can separate normal bearing signals from I2 and I3, but it is hard to tell normal ones from the I1 signals. There are a few differences in the WPE values of the normal bearing signal and the inner-race fault signals of three levels in Figure 12d. Figure 13 shows the distribution images of the test values of WPE of all the CD sequences of four-layer wavelet decomposition of 40 groups of the normal bearing signals and inner-race fault signals of different levels (I1, I2, and I3) by the same parameters m=5 and τ=2 for the test of the fifth fold simulation of five-fold cross-variation. The values of WPE of each layer of detailed coefficients show part of the characteristics of the signals in each subgraph. Therefore, we will use the WPE values of CA1, CA2, CA3, CA4, CD1, CD2, CD3, and CD4 to construct a vector to do the classification by ELM.

Figure 14 and Figure 15 show the distribution images of the values of PE of all the CA and CD sequences of four-layer wavelet decomposition based on ‘db3’ of 40 test groups of the normal bearing signals and inner-race fault signals of different levels (I1, I2, and I3) by the same parameters m=5 and τ=2 for the test of the fifth fold simulation of five-fold cross-variation. The PE values of the signals of CA or CD of a single layer of the normal bearing signals and inner-race fault signals of I1, I2, and I3 cannot be separated from each other.

Table 2 shows the training accuracy and testing accuracy of the classification methods with different detection parameters of normal bearing signals and inner-race fault signals of I1, I2, and I3 based on WPE and PE by four-layer wavelet decomposition based on ‘db3’ and ELM. K is the number of nodes in the hidden layer. Figure 16 shows the histogram of the training accuracy and testing accuracy. We divide the training and test sets for each signal in a ratio of 4:1, i.e., sets 1–40 are training sets, and sets 41–50 are test sets for each signal. For classification by using ELM, we first perform five-fold cross-validation to find the range of value of the number of hidden neurons K, then take the minimum value of the range of K as the number of hidden neurons, and the transfer function of the ELM is a sigmoidal function. When the number of hidden neurons is K-min, the training accuracy and testing accuracy of the classification method based on the methods listed in Table 2 is recorded by one simulation. In Table 2, we can see that the classification methods based on the feature vectors consisting of WPE (CA, CD), WPE (CD), or PE (CA, CD) have the best performances, the accuracies of them were 100%.

Figure 16 shows the maxima and minima of training and testing accuracy values of each kind of feature vector by the histogram of training accuracy and the testing accuracy of the classification methods with different detection parameters of normal bearing signals and inner-race fault signals of I1, I2, and I3 based on WPE and PE by four-layer wavelet decomposition based on ‘db3’ and ELM.

Figure 17 and Figure 18 show the distribution images of the values of WPE of all the CA and CD sequences of four-layer wavelet decomposition based on the ‘db3’ of 40 test groups of the normal bearing signals and inner-race fault signals of I1, I2, and I3 by the same parameters m=5 and τ=2. The WPE values of the signals of CA or CD of a single layer of the normal bearing signals and 3 types of fault signals of I1, O1, and B1 cannot be separated from each other.

Figure 19 and Figure 20 show the distribution images of the values of PE of all the CA and CD sequences of four-layer wavelet decomposition based on the ‘db3’ of 40 test groups of the normal bearing signals and inner-race fault signals of I1, I2, and I3 by the same parameters m=5 and τ=2. The PE values of the signals of CA or CD of a single layer of the normal bearing signals and three types of fault signals of I1, O1, and B1 cannot be separated from each other. Only in Figure 20a the PE values of the signals of CD1 can be separated from each other.

Table 3 shows the training accuracy and testing accuracy of the classification methods with different detection parameters of normal bearing signals and three types of fault signals of I1, O1, and B1 based on WPE and PE by four-layer wavelet decomposition based on ‘db3’ and ELM. K is the number of nodes in the hidden layer. Figure 21 shows the histogram of the training accuracy and testing accuracy. We divided the training and test sets for each signal in a ratio of 4:1, i.e., sets 1–40 are training sets, and sets 41–50 are test sets for each signal. For classification using ELM, first perform five-fold cross-validation to find the range of value of the number of hidden neurons K, then take the minimum value of the range of K as the number of hidden neurons, and the transfer function of the ELM is sigmoidal function. When the number of hidden neurons is K-min, the training accuracy and testing accuracy of the classification method based on the methods listed in Table 3 is recorded by one simulation. In Table 3, we can see that the classification methods based on the feature vectors of WPE (CA, CD), WPE (CD), PE (CA, CD), and PE (CD) have the best performances.

Figure 21 shows the maxima and minima of training and testing accuracy values of each kind of feature vector by the histogram of training accuracy and the testing accuracy of the classification methods with different detection parameters of normal bearing signals and inner-race fault signals of I1, O1, and B1 based on WPE and PE by four-layer wavelet decomposition based on ‘db3’ and ELM.

Table 4 shows the training accuracy and testing accuracy of the classification methods with different detection parameters of normal bearing signals and six types of fault signals of I1, I2, O1, O2, B1, and B2 based on WPE and PE by four-layer wavelet decomposition based on ‘db3’ and ELM. K is the number of nodes in the hidden layer. Figure 22 shows the histogram of the training accuracy and testing accuracy. We divided the training and test sets for each signal in a ratio of 4:1, i.e., sets 1–40 are training sets and sets 41–50 are test sets for each signal. For classification using ELM, firstly, we perform five-fold cross-validation to find the range of value of the number of hidden neurons K, then take the minimum value of the range of K as the number of hidden neurons, and the transfer function of the ELM is the sigmoidal function. In Table 4, we can see that the classification method based on the feature vector consisted of WPE (CA, CD), and has the best performance. When the number of hidden neurons is 37, the training accuracy and testing accuracy of the classification method based on WPE (CA, CD) can reach 98.57% by one simulation. At the same time, for the feature vector PE (CA, CD), the number of hidden neurons is 60, although the training accuracy can reach 98.21% and the testing accuracy is 97.14%. For the other feature vector by ELM listed in Table 4, the performances will be worse.

Figure 22 shows the maxima and minima of training and testing accuracy values of each kind of feature vector by the histogram of training accuracy and the testing accuracy of the classification methods with different detection parameters of normal bearing signals and inner-race fault signals of I1, I2, O1, O2, B1, and B2 based on WPE and PE by four-layer wavelet decomposition based on ‘db3’ and ELM.

As shown in Table 5, we compared the average runtime and average accuracy of the algorithm in this paper with the paper by Yang Y. In the paper of Yang Y., they use Normal, I1, I3, O1, O3, B1, and B3 to do the simulations based on the detrended fluctuation analysis (MFDFA) method-singularity power spectrum (SPS) method by ELM, denoted as MFDFA-SPS+ELM. The method of cross-validation for ELM in Yang Y.’s paper is random validation: firstly, randomly select 40 out of 50 sets of signals as training sets, and the other 10 sets are the test sets; secondly, take the number of hidden nodes to be 50; finally, the average of the accuracies obtained from five runs is taken as the average accuracy. In Table 5, we use Normal, I1, I2, O1, O2, B1, and B2 to do the calculations based on PE or WPE by wavelet decomposition and classified by the ELM with the random cross-validation used in the paper [15]. We also perform calculations using the same data (Normal, I1, I3, O1, O3, B1, and B3) as in Yang Y.’s paper and compare their average runtime and average accuracy using the methods in this paper. The difference in signal faults between levels 1 and 3 is relatively larger than the difference in faults between levels 1 and 2. We can see that the average accuracy obtained in Table 5 for different data types but with the same method. The average accuracy obtained for Type 1 is slightly less than the average accuracy obtained for Type 2 and the average runtime of the two data is close to each other. For the data from Type 2, we can see that the average accuracy of PE (CA, CD) +ELM reaches 96.57%, whereas the average accuracy of WPE (CA, CD) +ELM reaches 99.71%. The highest accuracy is 99.25% with the MFDFA-SPS method. Comparing the two methods, WPE (CA, CD) +ELM and MFDFA-SPS+ELM, with the same classification method and same parameters, it can be seen that the difference in average accuracy is 0.46%, but WPE (CA, CD) +ELM has a faster runtime.

## 5. Conclusions

In view of the non-stationary and nonlinear characteristics of rolling bearing vibration signals, this paper proposes a rolling bearing fault diagnosis and classification method based on weighted permutation entropy and permutation entropy by the wavelet decomposition and ELM. Firstly, the wavelet decomposition based on ‘db3’ is used to decompose the original vibration signals of rolling bearings into four layers of approximate components and detail components, and then the weighted permutation entropy of each layer of components is achieved to obtain the feature matrix. Finally, the obtained feature matrix is fed to the multi-classification ELM to realize the diagnosis and classification of normal bearing signals and six fault rolling bearing signals. The analysis of experimental data shows that the proposed methods using WPE (CA, CD), WPE (CD), and PE (CA, CD) have the same performances of the classification of normal bearing signals and inner-race fault signals of I1, I2, I3. The proposed method using WPE (CA, CD), WPE (CD), PE (CA, CD), and PE (CD) has the best classification performance of normal bearing signals and three types of fault states (I1, O1, B1). The proposed method using WPE (CA, CD) is the best one for multi-classification of normal bearing signals and six types of fault states (I1, I2, O1, O2, B1, B2) with 37–100 hidden neurons. In summary, although WPE and PE have similar accuracy in classifying a small number of classes of signals, WPE performs better in classifying multiple classes of signals, taking into account the effect of mutations.

We also compare the algorithm WPE (CA, CD) +ELM proposed in this paper with MFDFA-SPS+ELM and found that the algorithm in this paper has a faster runtime and the difference in the highest average accuracy is within 1.00% compared to the highest average accuracy of MFDFA-SPS+ELM. Both algorithms take into account the signal energy and have good performances. We compared the average accuracy of the two algorithms and found that they are equally effective and have good results in diagnosing and classifying rolling bearing signals. The MFDFA-SPS+ELM method should manually select the detection parameters by comparative analysis of the multifractal spectrum and singularity power spectrum first. However, our proposed WPE (CA, CD) +ELM method doesn’t have to select the detection parameters manually. We select the fixed WPE parameters of all of the CA and CD coefficients of four-layer wavelet decomposition based on the ‘db3’ of each signal as the detection feature vector. All the experiments are finished using MATLAB 2019 with intel CORE i7.

In the future, we will study more entropy algorithms, such as inverse sample entropy [19], phase transfer entropy [20], and symbol phase transfer entropy [21]. We will then attempt to make good use of them in order to analyze the bearing signals. We will translate the time series to the two-dimensional pictures [22] by Gramian Angular Summation Field (GASF) and Gramian Angular Difference Field (GADF), and then use the two-dimensional sample entropy method and two-dimensional multifractal methods to do the characteristic analysis of the signals.

## Figures and Tables

**Figure 1 entropy-24-01423-f001:**
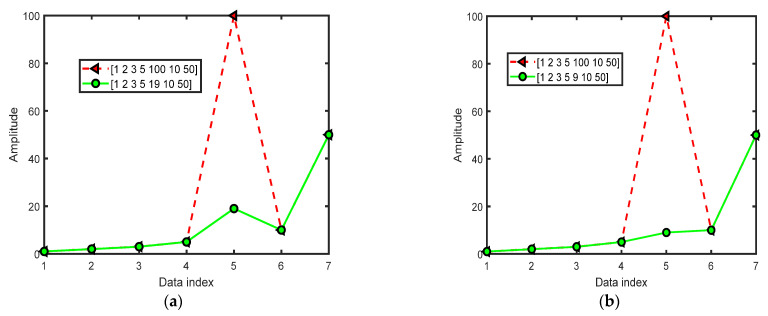
The data waveforms for the examples of PE and WPE. (**a**) The waveforms of $X_{1}=[1,2,3,5,100,10,50]$ and $X_{2}=[1,2,3,5,19,10,50]$; (**b**) The waveforms of $X_{1}=[1,2,3,5,100,10,50]$ and $X_{3}=[1,2,3,5,9,10,50]$.

**Figure 2 entropy-24-01423-f002:**
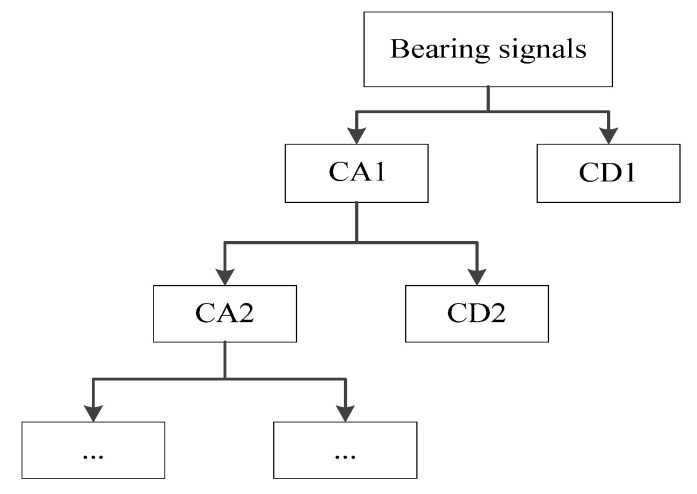
Wavelet decomposition diagram.

**Figure 3 entropy-24-01423-f003:**
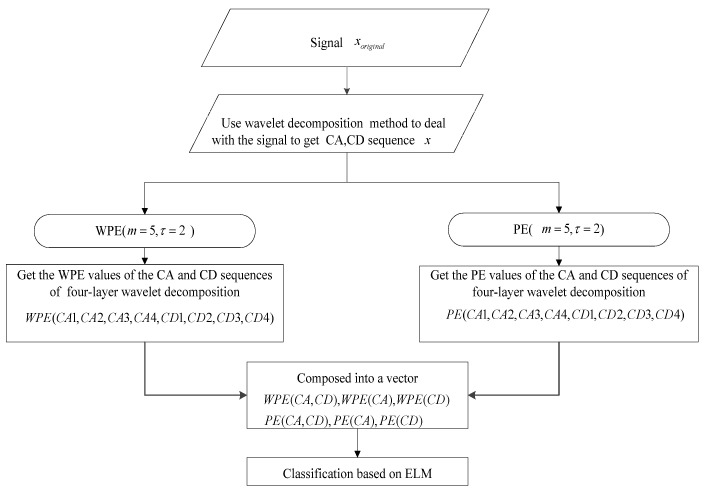
The flowchart of the proposed diagnostic method.

**Figure 4 entropy-24-01423-f004:**
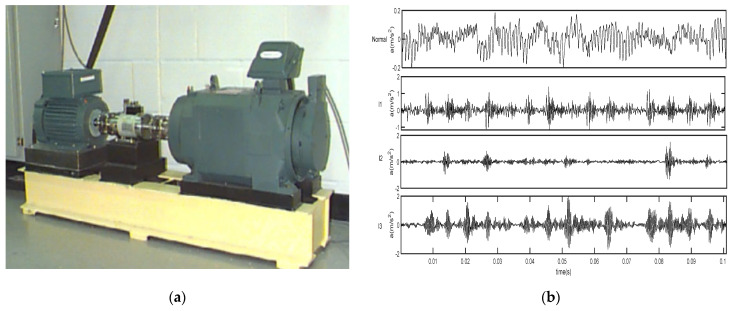
Experimental setup and the waveforms of four types of bearing signals. (**a**) The experimental setup; (**b**) the normal bearing signal and the inner-race fault of 3 levels I1, I2, and I3 corresponding to rolling bearing fault signals of 7 mils, 14 mils, and 21 mils.

**Figure 5 entropy-24-01423-f005:**
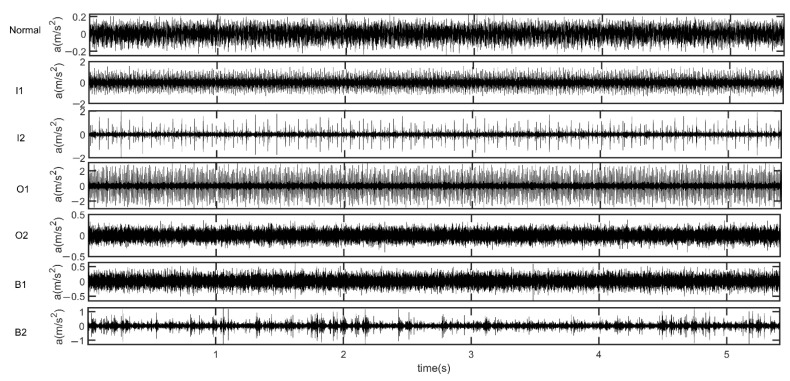
The waveforms of the bearing signals in seven different states.

**Figure 6 entropy-24-01423-f006:**
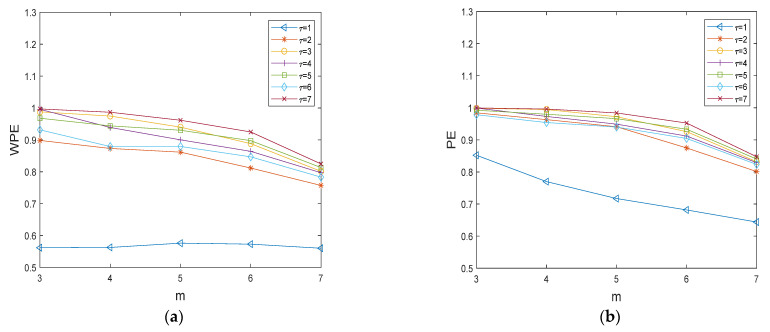
The WPE and PE curves of normal bearing signal with different time delays. (**a**) The WPE curves of normal bearing signal with different time delays; (**b**) The PE curves of normal bearing signal with different time delays.

**Figure 7 entropy-24-01423-f007:**
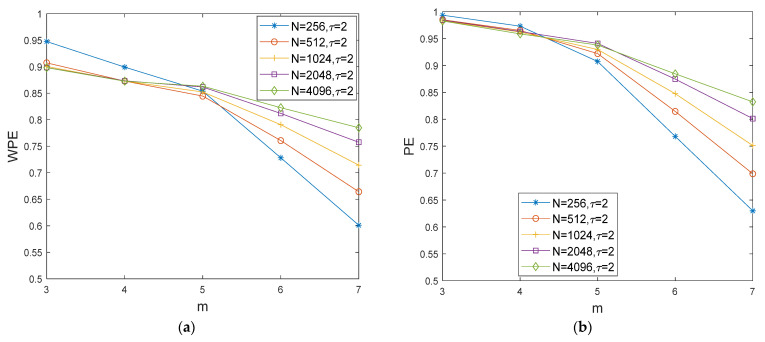
The WPE and PE curves of normal bearing signals with different N and different m. (**a**) The WPE curves of normal bearing signals with different N and different m; (**b**) the PE curves of normal bearing signals with different N and different m.

**Figure 8 entropy-24-01423-f008:**
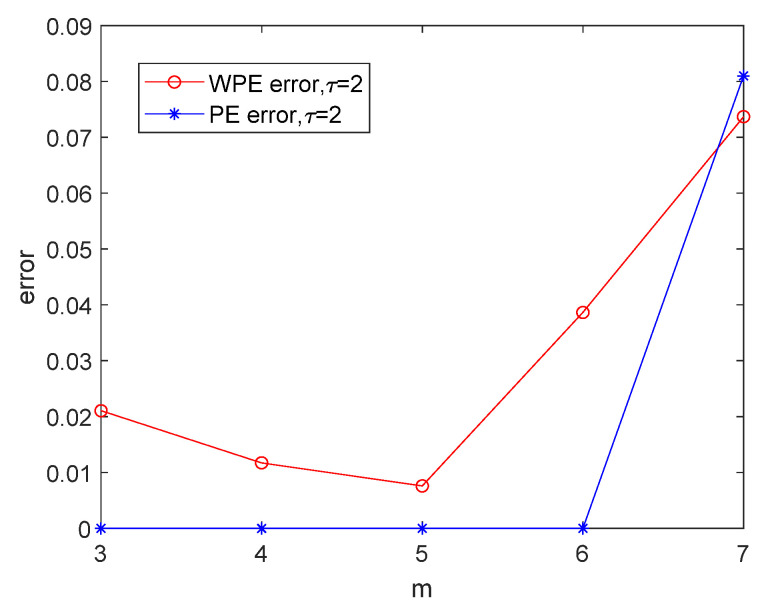
The error curves of WPE and PE of normal bearing signals in Figure 7.

**Figure 9 entropy-24-01423-f009:**
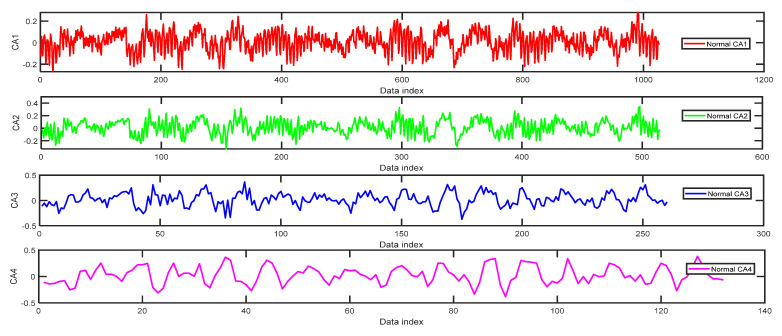
The approximation coefficient curves of four-layer wavelet decomposition based on ‘db3’ of normal bearing signal.

**Figure 10 entropy-24-01423-f010:**
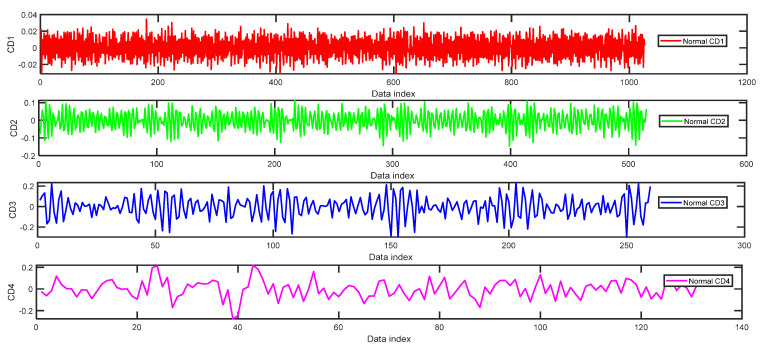
The detail coefficient curves of four-layer wavelet decomposition based on ‘db3’ of normal bearing signal.

**Figure 11 entropy-24-01423-f011:**
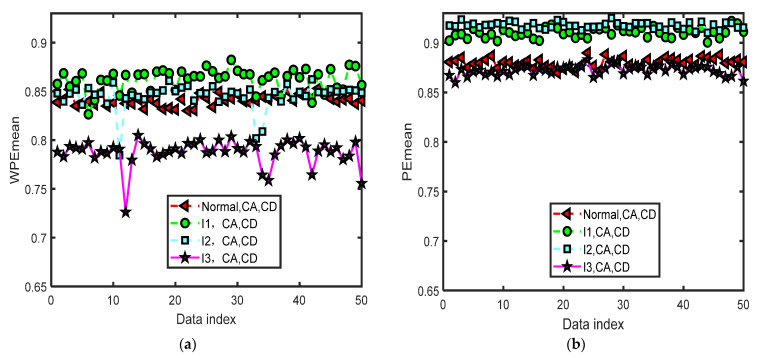
The curves of the mean values of WPE and PE of all the CA and CD sequences of four-layer wavelet decomposition of 200 groups of the normal bearing signals and inner-race fault signals of different levels (I1, I2, and I3) by the same parameters m=5 and τ=2. (**a**) The curves of the mean values of WPE; (**b**) the curves of the mean values of PE.

**Figure 12 entropy-24-01423-f012:**
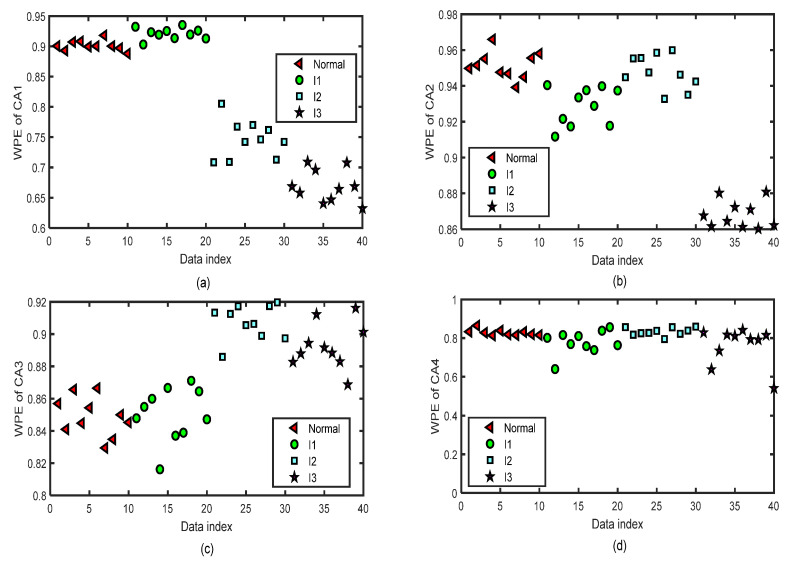
The distribution images of the values of WPE of all the CA sequences of four-layer wavelet decomposition based on ‘db3’ of 40 test groups of the normal bearing signals and inner-race fault signals of different levels (I1, I2, and I3) by the same parameters m=5 and τ=2 for the test of the fifth fold simulation of five-fold cross-variation. (**a**) WPE of CA1 after a four-layer wavelet decomposition of the signal. (**b**) WPE of CA2 after a four-layer wavelet decomposition of the signal. (**c**) WPE of CA3 after a four-layer wavelet decomposition of the signal. (**d**) WPE of CA4 after a four-layer wavelet decomposition of the signal.

**Figure 13 entropy-24-01423-f013:**
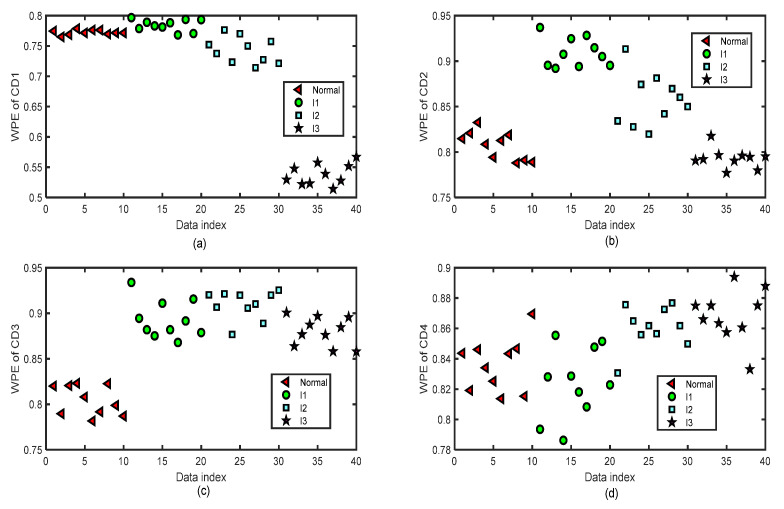
The distribution images of the values of WPE of all the CD sequences of four-layer wavelet decomposition based on ‘db3’ of 40 test groups of the normal bearing signals and inner-race fault signals of different levels (I1, I2, and I3) by the same parameters m=5 and τ=2 for the test of the fifth fold simulation of five-fold cross-variation. (**a**) WPE of CD1 after a four-layer wavelet decomposition of the signal. (**b**) WPE of CD2 after a four-layer wavelet decomposition of the signal. (**c**) WPE of CD3 after a four-layer wavelet decomposition of the signal. (**d**) WPE of CD4 after a four-layer wavelet decomposition of the signal.

**Figure 14 entropy-24-01423-f014:**
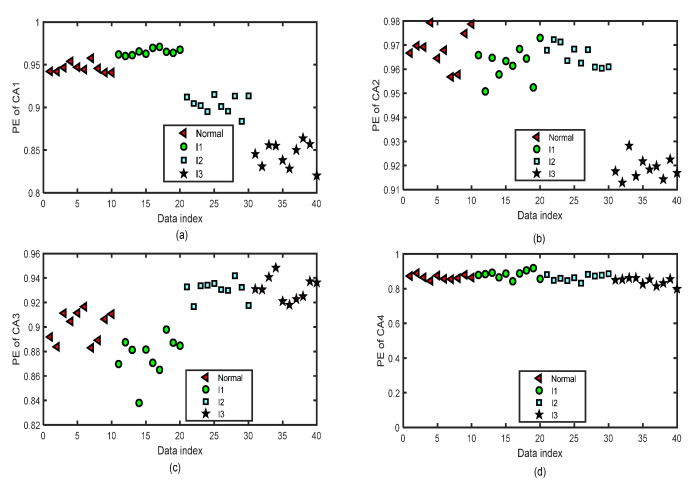
The distribution images of the values of PE of all the CA sequences of four-layer wavelet decomposition based on ‘db3’ of 40 groups of the normal bearing signals and inner-race fault signals of different levels (I1, I2, and I3) by the same parameters m=5 and τ=2 for the test of the fifth fold simulation of five-fold cross-variation. (**a**) PE of CA1 after a four-layer wavelet decomposition of the signal. (**b**) PE of CA2 after a four-layer wavelet decomposition of the signal. (**c**) PE of CA3 after a four-layer wavelet decomposition of the signal. (**d**) PE of CA4 after a four-layer wavelet decomposition of the signal.

**Figure 15 entropy-24-01423-f015:**
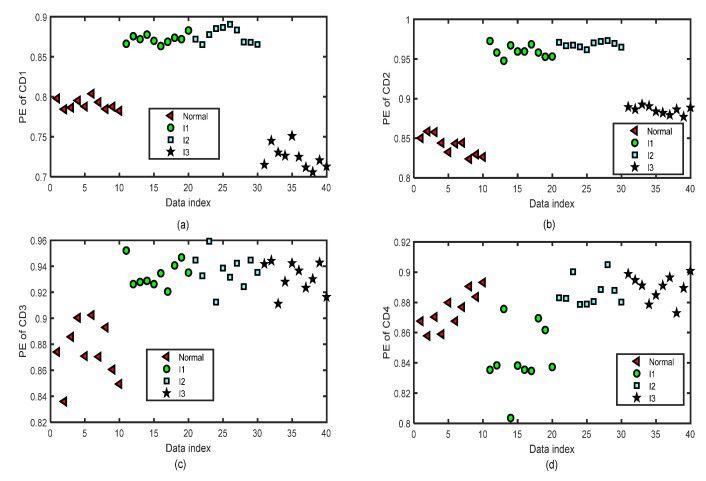
The distribution images of the values of PE of all the CD sequences of four-layer wavelet decomposition based on ‘db3’ of 40 groups of the normal bearing signals and inner-race fault signals of different levels (I1, I2, and I3) by the same parameters m=5 and τ=2 for the test of the fifth fold simulation of five-fold cross-variation. (**a**) PE of CD1 after a four-layer wavelet decomposition of the signal. (**b**) PE of CD2 after a four-layer wavelet decomposition of the signal. (**c**) PE of CD3 after a four-layer wavelet decomposition of the signal. (**d**) PE of CD4 after a four-layer wavelet decomposition of the signal.

**Figure 16 entropy-24-01423-f016:**
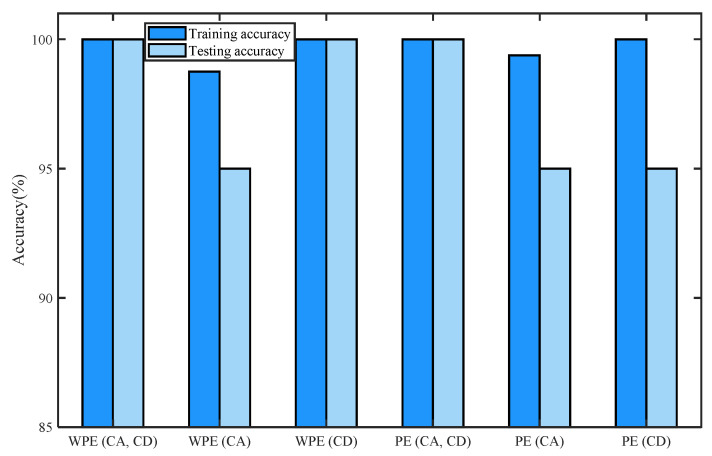
Histogram of training accuracy and the testing accuracy of the classification methods with different detection parameters of normal bearing signals and inner-race fault signals of I1, I2, and I3 based on WPE and PE by four-layer wavelet decomposition based on ‘db3’ and ELM.

**Figure 17 entropy-24-01423-f017:**
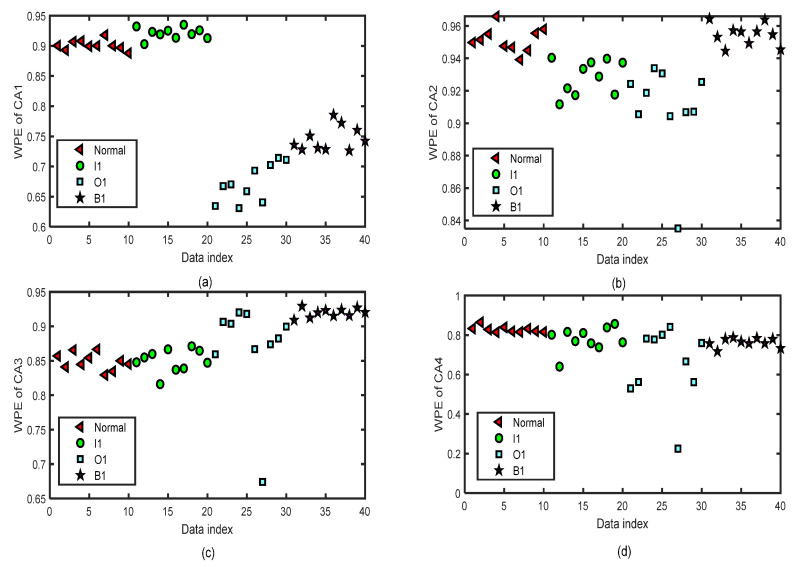
The distribution images of the values of WPE of all the CA sequences of four-layer wavelet decomposition based on ‘db3’ of 40 test groups of the normal bearing signals and three types of fault signals of I1, O1, and B1 by the same parameters m=5 and τ=2 for the test of the fifth fold simulation of five-fold cross-variation. (**a**) WPE of CA1 after a four-layer wavelet decomposition of the signal. (**b**) WPE of CA2 after a four-layer wavelet decomposition of the signal. (**c**) WPE of CA3 after a four-layer wavelet decomposition of the signal. (**d**) WPE of CA4 after a four-layer wavelet decomposition of the signal.

**Figure 18 entropy-24-01423-f018:**
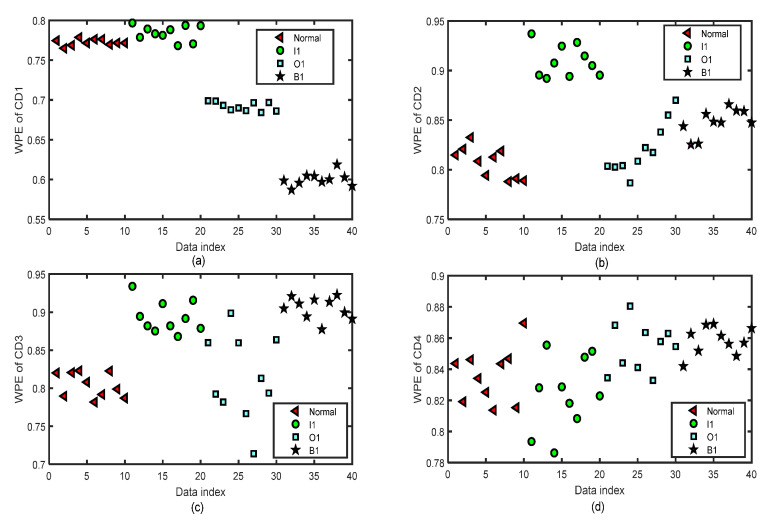
The distribution images of the values of WPE of all the CD sequences of four-layer wavelet decomposition based on ‘db3’ of 40 test groups of the normal bearing signals and three types of fault signals of I1, O1, and B1 by the same parameters m=5 and τ=2 for the test of the fifth fold simulation of five-fold cross-variation. (**a**) WPE of CD1 after a four-layer wavelet decomposition of the signal. (**b**) WPE of CD2 after a four-layer wavelet decomposition of the signal. (**c**) WPE of CD3 after a four-layer wavelet decomposition of the signal. (**d**) WPE of CD4 after a four-layer wavelet decomposition of the signal.

**Figure 19 entropy-24-01423-f019:**
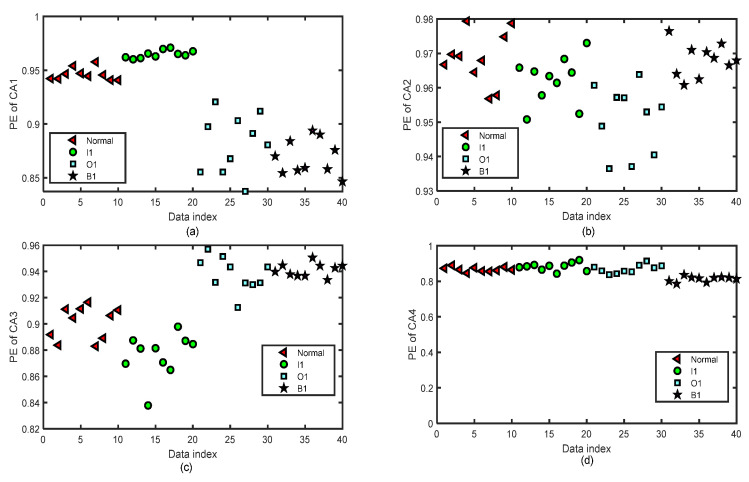
The distribution images of the values of PE of all the CA sequences of four-layer wavelet decomposition based on ‘db3’ of 40 test groups of the normal bearing signals and three types of fault signals of different levels (I1, O1, and B1) by the same parameters m=5 and τ=2 for the test of the fifth fold simulation of five-fold cross-variation. (**a**) PE of CA1 after a four-layer wavelet decomposition of the signal. (**b**) PE of CA2 after a four-layer wavelet decomposition of the signal. (**c**) PE of CA3 after a four-layer wavelet decomposition of the signal. (**d**) PE of CA4 after a four-layer wavelet decomposition of the signal.

**Figure 20 entropy-24-01423-f020:**
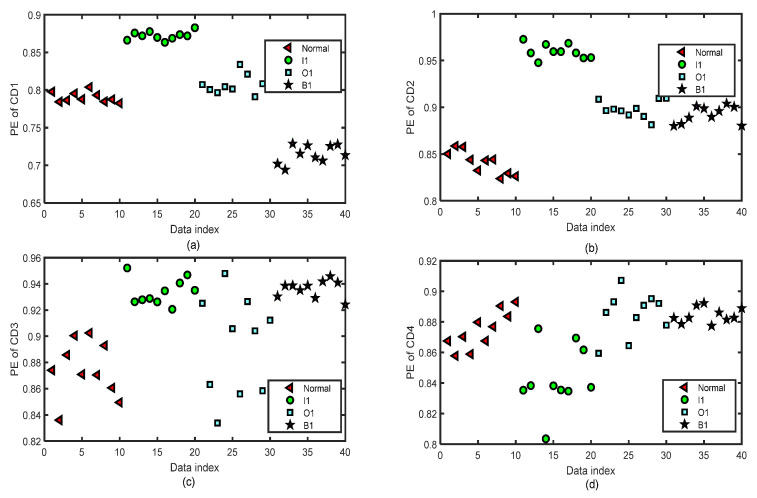
The distribution images of the values of PE of all the CD sequences of four-layer wavelet decomposition based on ‘db3’ of 40 test groups of the normal bearing signals and three types of fault signals of I1, O1, and B1 by the same parameters m=5 and τ=2 for the test of the fifth fold simulation of five-fold cross-variation. (**a**) PE of CD1 after a four-layer wavelet decomposition of the signal. (**b**) PE of CD2 after a four-layer wavelet decomposition of the signal. (**c**) PE of CD3 after a four-layer wavelet decomposition of the signal. (**d**) PE of CD4 after a four-layer wavelet decomposition of the signal.

**Figure 21 entropy-24-01423-f021:**
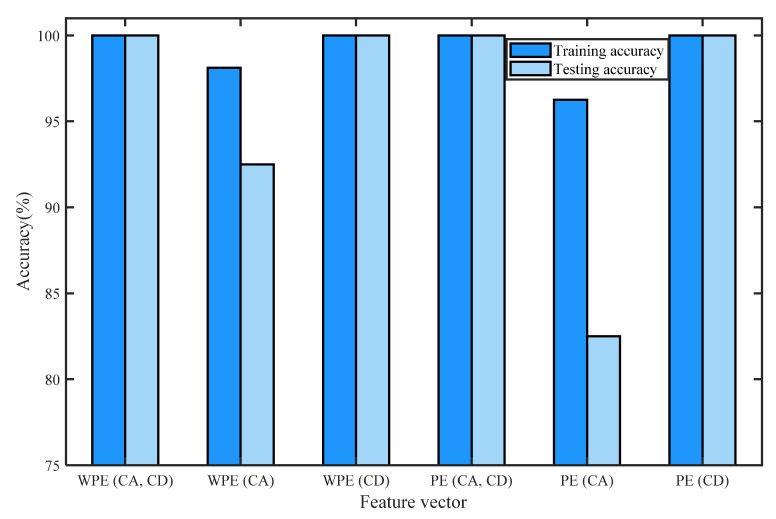
Histogram of training accuracy and the testing accuracy of the classification methods with different detection parameters of normal bearing signals and three types of fault signals of I1, O1, and B1 based on WPE and PE by four-layer wavelet decomposition based on ‘db3’ and ELM.

**Figure 22 entropy-24-01423-f022:**
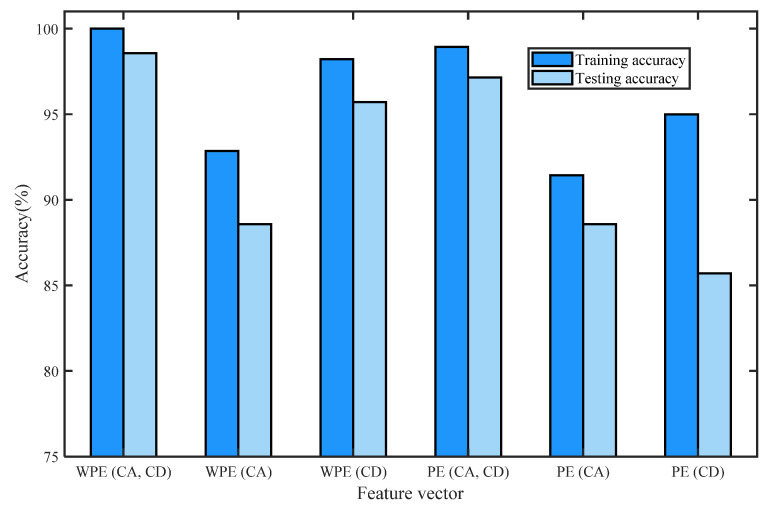
Histogram of training accuracy and the testing accuracy of the classification methods with different detection parameters of normal bearing signals and six types of fault signals of I1, I2, O1, O2, B1, and B2 based on WPE and PE by four-layer wavelet decomposition based on ‘db3’ and ELM.

**Table 1 entropy-24-01423-t001:** Accuracy of the training set obtained with different decomposition layers of different wavelet basis functions under both PE and WPE algorithms.

Basis	PE, WPE	N = 4		N = 5		N = 6		N = 7		N = 8	
		PM	K	PM	K	PM	K	PM	K	PM	K
Db3	PE	99.14%	48–100	99.36%	82–100	98.93%	56–100	98.86%	80–100	98.79%	70–100
	WPE	100.00%	18–100	100.00%	27–100	100.00%	36–100	100.00%	44–100	100.00%	45–100
Db4	PE	98.79%	48–100	99.07%	33–100	99.29%	52–100	99.29%	43–100	99.21%	64–100
	WPE	100.00%	17–100	100.00%	23–100	100.00%	36–100	100.00%	43–100	100.00%	45–100
Db5	PE	99.14%	67–100	98.86%	40–100	99.07%	48–100	99.14%	43–100	99.00%	72–100
	WPE	100.00%	29–100	100.00%	39–100	100.00%	35–100	100.00%	66–100	100.00%	73–100
Db6	PE	98.64%	36–96	99.50%	60–100	99.36%	62–100	99.43%	61–100	99.50%	80–100
	WPE	100.00%	21–100	100.00%	33–100	100.00%	38–100	100.00%	44–100	100.00%	46–100
Db7	PE	98.29%	87–100	98.93%	73–100	99.07%	57–99	99.64%	78–99	99.64%	73–100
	WPE	100.00%	15–100	100.00%	24–100	100.00%	38–100	100.00%	43–100	100.00%	35–100

**Table 2 entropy-24-01423-t002:** The training accuracy and the testing accuracy of the classification methods with different detection parameters of normal bearing signals and inner-race fault signals of I1, I2, and I3 based on WPE and PE by four-layer wavelet decomposition based on ‘db3’ and ELM.

	Feature Vector	Training Accuracy	Testing Accuracy	K-Range	K-min
1	WPE (CA, CD)	100.00% (160/160)	100.00% (40/40)	9–100	9
2	WPE (CA)	98.75% (158/160)	95.00% (38/40)	21–100	21
3	WPE (CD)	100.00% (160/160)	100.00% (40/40)	28–100	28
4	PE (CA, CD)	100.00% (160/160)	100.00% (40/40)	9–100	9
5	PE (CA)	99.38% (159/160)	95.00% (38/40)	32–100	32
6	PE (CD)	100.00% (160/160)	95.00% (38/40)	35–100	35

**Table 3 entropy-24-01423-t003:** The training accuracy and the testing accuracy of the classification methods with different detection parameters of normal bearing signals and three types of fault signals of I1, O1, and B1 based on WPE and PE by four-layer wavelet decomposition based on ‘db3’ and ELM.

	Feature Vector	Training Accuracy	Testing Accuracy	K-Range	K-min
1	WPE (CA, CD)	100.00% (160/160)	100.00% (40/40)	8–100	8
2	WPE (CA)	98.12% (157/160)	92.50% (37/40)	20–100	20
3	WPE (CD)	100.00% (160/160)	100.00% (40/40)	25–100	25
4	PE (CA, CD)	100.00% (160/160)	100.00% (40/40)	9–100	9
5	PE (CA)	96.25% (154/160)	82.50% (33/40)	85–100	85
6	PE (CD)	100.00% (160/160)	100.00% (40/40)	13–100	13

**Table 4 entropy-24-01423-t004:** The training accuracy and the testing accuracy of the classification methods with different detection parameters of normal bearing signals and six types of fault signals of I1, I2, O1, O2, B1, and B2 based on WPE and PE by four-layer wavelet decomposition based on ‘db3’ and ELM.

	Feature Vector	Training Accuracy	Testing Accuracy	K-Range	K-min
1	WPE (CA, CD)	100.00% (280/280)	98.57% (69/70)	37–100	37
2	WPE (CA)	92.86% (260/280)	88.57% (62/70)	75–100	75
3	WPE (CD)	98.21% (275/280)	95.71% (67/70)	39–100	39
4	PE (CA, CD)	98.93% (277/280)	97.14% (68/70)	60–100	60
5	PE (CA)	91.43% (256/280)	88.57% (62/70)	70–100	70
6	PE (CD)	95.00% (266/280)	85.71% (60/70)	83–100	83

**Table 5 entropy-24-01423-t005:** Comparison of runtime and average accuracy.

Data Types	Algorithm	Average Runtime (s)	Average Accuracy (%)
Type 1: Normal, I1, I2, O1, O2, B1, B2	PE (CA, CD) + ELM	3.90	97.43
	PE (CA, CD) + SVM	3.04	94.29
	WPE (CA, CD) + ELM	3.85	99.14
	WPE (CA, CD) + SVM	2.60	98.57
Type 2: Normal, I1, I3, O1, O3, B1, B3	PE (CA, CD) + ELM	3.94	96.57
	PE (CA, CD) + SVM	7.30	95.71
	WPE (CA, CD) + ELM	3.89	99.71
	WPE (CA, CD) + SVM	5.89	98.57
	MFDFA + ELM	54.41	92.46
	SPS + ELM	2.05	80.83
	MFDFA-SPS + ELM	54.56	99.25
	MFDFA-SPS + SVM	54.22	95.71
	MFDFA-SPS + LSSVM	55.52	95.00

## Data Availability

All data used in the experiments can be downloaded from the links: https://www.cnblogs.com/gshang/p/10712809.html (accessed on 20 November 2021).

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
