# Peer review of "Fault Diagnosis of Rolling Bearings Based on WPE by Wavelet Decomposition and ELM"

_entropy, 2022, doi:10.3390/e24101423_

Round 1

Reviewer 1 Report

The article describes the activities of researchers related to the assessment of the suitability of individual methods of processing diagnostic signals from rolling bearings. The authors indicated a specific percentage match of individual methods. This approach is interesting, but the conclusions are not sufficient, and the introduction is too general and short.

1 The sources [4] [5] [6] are not well described. It should be [4,5,6]. Likewise, other sources follow.

2. The introduction is too poor. It does not describe the complexity of the problem of bearing diagnosis and the problems associated with the obtained diagnostic signals.

3. Drawings, especially those related to waveforms, are of poor quality.

4 Why is it that researchers use the units "mil"? Is this only due to the data obtained?

5. How many bearings have been tested? For signals to be taken from different repetitions?

6. Why do the authors provide different numbers of significant digits for percentage differences? Why are there so many of them?

7. The conclusions should also add whether these "inferior" methods are useful for and diagnosing bearings. Or are they also sufficient? Or what direct benefits are there in the methods that the authors consider to be better?

Reviewer 2 Report

In this work, a machine learning scheme is presented for bearing fault detection. In general, the wavelet decomposition and weighted permutation entropy methods are used. Although promising results are obtained, there are several issues that have to be addressed and clarified.

It is not clear the contribution of your work. Is your work only another solution for a solved problem?

It is not clear why the Authors refer to bearings in wind turbines when the database from the Bearing Data Center has been used in many reported works  by considering only the bearing fault concept (no wind turbines). In fact, it seems that the focus of wind turbines in the introduction section and title is used to indicate novelty or a current application; however, the database does not have such focus.  Please discuss in detail.

Include the computational time that is mentioned in line 318.

During training and testing, there was k-fold validation? Please consider it.

Discuss the impact of the mother wavelet and justify the level of decomposition.

Provide more data to be able to reproducing the results of Tables 2 y 3. Also add a plot to see maxima and minima values.

For the classification effectiveness provide the confusion matrix, as well as the ROC, precision, recall, etc.

Also, it is no clear how do you determine the structure of the ELM. Is it the optimal scheme?

In the method several values are used, but they are not justified. What is the impact of using other values? Are they the best ones for your method

As the treated problem has been researched by many authors, a comparison (qualitative and quantitative, e.g., a table) has to be included and discussed in order to highlight the contributions and improvements.

Please increase quality for all the figures (

Round 2

Reviewer 1 Report

I thank the authors for comprehensive answers to my review.

Reviewer 2 Report

All the comments and suggestions have been properly addressed. This Reviewer recommends the manuscript acceptance. 

Please only check that all the information presented in the Response document is included in the manuscript (e.g. the confusion matrices).